# Satisfaction with Life: Mediating Role in the Relationship between Depressive Symptoms and Coping Mechanisms

**DOI:** 10.3390/healthcare9070787

**Published:** 2021-06-23

**Authors:** Daniela Almeida, Diogo Monteiro, Filipe Rodrigues

**Affiliations:** 1Sports Science School of Rio Maior, Polytechnique Institute of Santarém (ESDRM-IPSantarém), 2040-413 Rio Maior, Portugal; danivila76@hotmail.com; 2ESECS, Polytechnique of Leiria, 2411-901 Leiria, Portugal; diogo.monteiro@ipleiria.pt; 3Research Centre in Sports, Health and Human Development (CIDESD), 5001-801 Vila Real, Portugal; 4Life Quality Research Center (CIEQV), 2040-413 Rio Maior, Portugal

**Keywords:** adaptive coping, maladaptive coping, life satisfaction, depressive symptoms, Portuguese population

## Abstract

The purpose of this study was to analyze the mediating role of life satisfaction in the relationship between fourteen coping strategies and depressive symptoms in the Portuguese population. To undertake this work, 313 Portuguese adults aged 18 to 70 years (M = 30.73; SD = 10.79) were invited to participate in this study. Their participation was completely voluntary, and participants granted and signed informed consent previously to the filling of the validated Portuguese questionnaires. These questionnaires measured depressive symptoms, coping, and life satisfaction. The results revealed that life satisfaction displayed a mediating role in the relationship between adaptive coping mechanisms, specifically between active coping, planning, reinterpretation, and acceptance and depressive symptoms, showing a negative and significant indirect effect. Maladaptive coping mechanisms of self-blame, denial, self-distraction, disengagement, and substance use had a significant positive association with depressive symptoms, considering the mediating role of satisfaction with life. Current investigation provides initial evidence of how each coping mechanism is associated with satisfaction with life and depressive symptoms. This study clearly demonstrates that not all coping strategies are capable of influencing well-being indicators and that health professionals should focus on endorsing those that are significantly associated with lowering depressive symptoms and increasing overall satisfaction with life.

## 1. Introduction

In 2015, the World Health Organization [WHO] [1] had estimated that more than 300 million people suffered from depression, which represented 4.4% of the world population. The WHO [1] further stated that this number would increase throughout the following years. By the end of the 90s, the evidence available [1] on the impact of mental diseases revealed that it was urgent to make mental health one of the top priorities in the public health agenda, both nationally and internationally. It also underlined that only the development of new knowledge about the nature, causes, and consequences of mental disorders and a new understanding of the effectiveness of the interventions and mental health services could pave the way to new hope regarding the resolution of this disorder [2].

### 1.1. The Concept of Depression: Definition and Symptoms

The WHO [1] defines depression as a common mental disorder. Kessler [2] adds that, beyond common, it is also a disorder severely harmful and recurrent, and it is highly prevalent in the entire world. Depression represents a state of deep dismay, marked by apathy, negativity, and behavioral inhibition [3], compromising the daily functioning and well-being of an individual [4]. In a study developed by Wang et al. [5], on top of sadness, other symptoms of the disease were identified, such as reduced ability to process thoughts, reduced mental activity, cognitive dysfunction, and physical symptoms. The DSM-5 [6] stipulates nine criteria for depression, of which at least five must be present (see Table 1).

The symptoms associated with depression affect one’s ability to function at work or school and to deal with daily life events [1]. Thus, this disorder has a marked impact in the quality of life and goes beyond the disease itself: there is an increase in mortality due to suicide, cardiovascular, or cerebrovascular disease, as individuals with depression tend to have reduced life expectancy when compared with the general population [7]; there is psychosocial incapacity; reduced productivity in work and day-to-day activities; and higher risk of work absenteeism, all of which may also lead to problems within the household and with family members, potentially ending in couple separation or divorce in a moment when the individual most needs social support [8].

Literature has provided evidence that coping mechanisms are possible strategies for the control and adaptation of depressive symptoms, potentially even reducing their effect in depression [9]. Indeed, studies describe coping mechanisms as potentiators in the reduction of depressive symptoms, as they can be considered key variables in promoting mental health and reducing the risk of an individual being diagnosed with depression [10,11,12]. Therefore, it is important that research on coping strategies is continued as a means to provide clear evidence on how to create clinical interventions to reduce depressive symptoms and prevent the development of depressive episodes.

### 1.2. Coping Mechanisms

Carver et al. [13] conceptualized coping as the process of executing an answer to a potential threat. It generally refers to a cognitive and behavioral response to negative external events whereby if coping is effective, the individual is able to solve the problem or reduce the associated negative emotion, overcoming the stress barrier [13]. An adequate coping answer can lead one to reassess a threat as less threatening [12]. The word “coping” relates to “facing”, “dealing with”, or “adapting to” and may be defined as the effort undertaken to face a certain difficult situation and reduce the associated stress levels [14].

According to Carver [15], coping can assume 15 differentiating forms with different implications: active coping (e.g., adopting specific measures to solve the problem); planning (e.g., defining a strategy to deal with the situation); suppression of competing activities (e.g., focusing on the problem and, if required, leaving behind other actions and activities); positive reinterpretation and learning (e.g., trying to face the situation from a different and positive perspective); acceptance (e.g., learning to live with the situation); searching for instrumental social support (e.g., speaking with someone that can provide helpful information or help understand the situation); search of emotional social support (e.g., speaking with someone about what one is feeling); focus on and venting emotion (e.g., unloading emotions to feel relieved); behavioral disengagement (e.g., admitting one cannot deal with the problem and ceasing to try); mental disengagement (e.g., thinking of something else other than the problem); denial (e.g., telling oneself “*this is not happening*”); substance use (e.g., using alcohol to relax and abstract), humor (e.g., laughing about the situation), restraint coping (e.g., waiting for the right moment to act) and, finally, religious coping (e.g., resorting to religion in stressful situations). As a way to aggregate processes of assessing coping, considering theoretic conceptualizations, some authors [16,17,18] have grouped the coping mechanisms proposed by Carver [15] into adaptive coping [i.e., active coping, acceptance, humor, religious coping, planning, reinterpretation, instrumental support, and emotional support] and maladaptive coping (i.e., disengagement, denial, focus on and venting emotions, self-blame, self-distraction and substance use).

Adaptive coping strategies tend to be associated with desirable results (e.g., high levels of satisfaction with life; absence of depressive symptoms), while maladaptive coping strategies tend to be associated to undesirable results (e.g., depressive symptoms), as described by Su et al. [16]. Holubova et al. [19] differentiates these strategies as positive (adaptive) or negative (maladaptive), affirming that adaptive coping is considered effective and maladaptive coping reflects an inability to handle stressful events. According to Teques et al. [18], adaptive coping is positively associated to the regulation of emotion unlike what takes place with maladaptive strategies, which are negatively associated with such regulation.

### 1.3. Satisfaction with Life: Potential Mediator between Coping Mechanisms and Depressive Symptoms

Well-being can be defined in two main components: the emotional or affective component and the critical or cognitive component [19,20]. The cognitive component has also been conceptualized as satisfaction with life, which corresponds to a critical and cognitive assessment of one’s own life; thus, it can be indirectly influenced by the emotional component but is not, in itself, a direct measure of emotion [19]. Therefore, the assessment on how satisfied one is with their current life state is based on a comparison with a pattern that each individual defines for themselves and is not externally imposed on the individual [19].

Satisfaction with life is aligned according to the way in which one individual adjusts to the context that they are in, which has a connection with coping strategies. Tran and Chantagul [3] demonstrated that using adaptive coping strategies is positively and significantly associated with the level of satisfaction with life. On the other hand, maladaptive coping mechanisms are negatively and significantly associated with the level of satisfaction with life. This study concludes that the more adaptive coping mechanisms are used to deal with stressful events, the higher the level of satisfaction with life. On the contrary, the higher the use of maladaptive coping strategies, the lower the level of satisfaction with life [3]. According to Yang et al. [21], satisfaction with life is a resource that includes autonomy, control, beliefs, positive emotions, emotional regulation, problem-solving, adaptation, and balance throughout the life cycle. Therefore, it is expected that adaptive coping mechanisms have a positive correlation with satisfaction with life.

A study by Moksnes et al. [22] showed that high satisfaction with life is associated with a series of positive personal, behavioral, psychological, and social results, but low satisfaction with life is associated with higher levels of stress, psychologic disorders, and behavioral problems. Previous research done with adult populations reinforces this significant inverse relationship between satisfaction with life and depression [23,24,25].

McKnight et al. [26] demonstrated that satisfaction with life works as a mediator in the relation between the most stressful events in individuals lives, internalization symptoms (i.e., depression and anxiety), and externalization symptoms (i.e., aggressive and delinquent behavior). Moksnes et al. [27] supports this assumption, as it is theoretically expected that satisfaction with life may play a similar role regarding connection of the coping mechanisms and depressive symptoms. In other words, considering that there is a negative connection between adaptive coping mechanisms and depressive symptoms, it is hypothesized that satisfaction with life may work as a mediator, in that higher levels of adaptive coping and satisfaction with life are associated with lower depressive symptoms.

Moksnes et al. [27] demonstrated a positive significant relation between school stress and depressive symptoms and a negative significant relation between school stress and satisfaction with life. As for the potential mediating role of satisfaction with life, they verified that this was a partial mediator between stress and depressive symptoms. Studies reflect the complexity of the interaction between the stress one individual experiences regarding their school performance and mental health and the role of satisfaction with life as a potentially important mediator in that relationship [27]. These authors offer evidence of a significant connection between satisfaction with life and depressive symptoms as well as the mediating role that satisfaction with life may play in this particular relation. Satisfaction with life was a partial mediator in the relationship between stress and depressive symptoms. The results are supported by a related study that showed that satisfaction with life mediated the relationship between stressful life events and internalization symptoms [26]. On top of that, it is postulated that individuals with higher satisfaction with life will have more overall confidence if the resources they need to cope with stressful events are available to them [27]. Individuals using adaptive coping mechanisms are more likely to experience high levels of satisfaction with life and lower levels of depressive symptoms [22,27].

### 1.4. Current Study

Although there is some evidence that coping may act as an adaptation strategy to a possible depressive symptom, studies are scarce that have analyzed the mediating role of satisfaction with life and how it may relate to the different coping mechanisms and depressive symptoms. Moksnes et al. [27] is one of the few studies that researched satisfaction with life as a potential mediator between school-related stress and depressive symptoms, an indicator that more research is necessary. However, it is important to notice that, in the study of Moksnes et al. [22], the fourteen coping mechanisms were considered globally, leaving us with limitations regarding the understanding of the associative effect of each coping mechanism with depressive symptoms. Indeed, Tran and Chantagul [3] mentioned the lack of knowledge between each coping strategy, depressive symptoms, anxiety, and satisfaction with life, as there are not many studies that demonstrate a relationship between these variables. A deep study about the relationship between each coping mechanism, satisfaction with life, and depressive symptoms may offer essential knowledge to create tools to promote better well-being levels and lower negative psychologic indicators (e.g., stress, anxiety), considering the potential coping mechanisms, essential for well-being in the adult population.

With this in consideration, in the face of varied factors that may contribute to the development of mental disorders (such as depression) as well as their consequences for the well-being and quality of life of the individuals, it is important to study the relationship between coping mechanisms, depressive symptoms, and satisfaction with life. Therefore, the objective of this study was to analyze the mediating role of satisfaction with life with each of the fourteen coping mechanisms and with depressive symptoms.

According to the literature, it is speculated that: (i) adaptive coping mechanisms tend to help people adapt better to stressful situations and present fewer depressive symptoms, opposite to those who use maladaptive mechanisms [15]; (ii) the higher the levels of adaptive coping mechanisms, the better the level of satisfaction with life. On the other hand, the more maladaptive coping mechanisms used, the lower the levels of satisfaction with life. Thus, coping mechanisms are directly and significantly related to the levels of satisfaction with life [3]; (iii) there is evidence of a negative significant relationship between satisfaction with life and depressive symptoms [27]; and (iv) it is expected from a theoretical point of view that satisfaction with life may have a mediating role regarding the relationship between coping mechanisms and depressive symptoms [21]. Considering a negative relationship between adaptive coping mechanisms and satisfaction with life with depressive symptoms, it is hypothesized that satisfaction with life may act as a mediator. It is also hypothesized that maladaptive coping mechanisms are positively correlated with depressive symptoms and show an indirect positive and significant relation with satisfaction with life.

## 2. Materials and Methods

### 2.1. Design and Participants

This study had a cross-sectional design in order to analyze the different relationships between the variables being assessed. To correctly perform the study, Portuguese individuals were randomly invited to participate voluntarily, according to the following inclusion criteria: all participants were at least 18 years of age and completed the questionnaire voluntarily and anonymously. Excluding criteria were: any participant that did not declare informed consent and that did not answer the complete questionnaire (missing values above 5%). The final sample comprised 313 participants (184 female and 129 male), with ages between 18 years old and 70 years old (M = 30.73; SD = 10.79).

### 2.2. Procedures

Collection of information for this project was for strictly scientific studies, which is the reason why, as to the ethic code of conduct, confidentiality was guaranteed, and no information was transmitted individually to third parties. In this way, the data collection process was done voluntarily according to the guiding principles described in the Helsinki Declaration. After the study was approved by the Ethical Commission Board (ref: CE/IPLEIRIA/17/2021), the questionnaire was built using three validated instruments for the Portuguese population. A non-probabilistic sampling technique to collect data was used; specifically, data were collected from a convenience sample of the population. The participants had access to the questionnaire online using Google Forms created for the study and promoted using digital media (e.g., social networks, academic e-platforms). It is important to make note that no information regarding name, address, e-mail, or any other personal information was collected.

### 2.3. Instruments

Every participant filled in a set of questionnaires that were composed by sociodemographic characterization in terms of gender and age. Following that, the participants filled out the following questionnaires:

[i] Brief Cope, Portuguese version [14]. This was used to measure the 14 coping mechanisms proposed by Carver. This instrument has 28 questions assessing coping mechanisms (2 items for each mechanism; e.g., “*I pray or meditate*.”; “*I turn to alcohol or other drugs [pills, etc.] to make me feel better*.”), and participants answered each question using a Likert scale going from 1 (*Never did this*) to 5 (*Always do this*).

[ii] Satisfaction with Life Scale, Portuguese version [28]. This was used to measure the degree of satisfaction with life in general terms. This questionnaire has 5 items (e.g., “*I’m happy with my life*.”) to which participants answered using a Likert scale going from 1 (*Strongly disagree*) to 5 (*Strongly agree*).

[iii] Beck Depression Inventory [29], Portuguese version [30]. This was used to assess depressive symptoms. This questionnaire has 21 groups, each with 4 options for answering, that refer to individual states of being. As each participant reads through the options in each group, they have to select the answer that best describes how they feel (e.g., Group I— “*I don’t feel sad*”; “*I feel sad*.”; “*I’m always sad and I can’t avoid it*.”; “*I’m so sad or miserable that I can’t take this anymore*.”).

### 2.4. Statistical Analysis

Initially, a descriptive analysis of the variables was done using the software IBM SPSS Statistics v23 to obtain the average and standard deviation, followed by an analysis of the normal distribution (skewness and kurtosis). The following data for normal distribution were considered: skewness between −2 and +2 and kurtosis values between −7 and +7. Additionally, an analysis of bivariate correlations was done, specifically Pearson correlations.

In order to obtain answers relevant to the objectives of this studies, structural equation modeling were used based on Hair et al. [31]. This type of statistical analysis was considered, as it capable of producing direct and indirect effects [31]. Additionally, structural equation modeling is a modern statistical method that allows one to evaluate causal hypotheses on a set of intercorrelated, nonexperimental data, which is the case of this research. Chi-square (χ^2^) and corresponding degrees of freedom (df) are reported for transparency, but were not examined to assess model adequacy, as they are subject to the size of the sample and model specifications [31]. First, an analysis of structural equation modeling was performed by the maximum likelihood method using the software AMOS 23.0. For a detailed analysis, 14 structural equation models were performed, considering each of the mechanisms as an independent variable. Depressive symptoms were introduced in the system as a dependent variable and calculated using average, as proposed by Beck et al. [30]. The models were analyzed using traditional adjustment and incremental values described by several authors [31,32], specifically using the following adjustment indexes: comparative fit index (CFI), Tucker–Lewis index (TLI), standard root mean residual (SRMR), and root mean square error of approximation (RMSEA), with its respective confidence interval of 90% (IC 90%). For the previously referred indexes, the following cutoff values were considered acceptable: CFI and TLI ≥ 0.90 and SRMR and RMSEA ≤ 0.8, as proposed by several authors [31,32].

Secondly, direct and indirect effects were analyzed according to standardized beta coefficients (β). The significance of the standardized coefficients for direct and indirect effects was measured with a confidence interval (IC) of 95%, being considered significant when IC was not encompassing the value of 0 [33].

### 2.5. Sample Size

Sample size for multivariate analyzes was performed according to the recommendations of Westland [34], using the online calculator by Soper [35]. We included the following parameters: anticipated effect size (0.2); desired statistical power level (0.8); number of latent variables (2); number of observed variables (1); and probability level (0.05). Considering these parameters, the minimum required sample was 223, which was respected in the present study.

## 3. Results

### 3.1. Preliminary Results

Data from all 313 participants was inputted, as none presented missing values due to the way the online questionnaire was constructed. In Table 2, it is possible to observe from a descriptive perspective that adaptive coping mechanisms present a higher average compared to maladaptive coping mechanisms. Specifically, the variable “planning” has the highest average, following by the mechanisms “active coping” and “reinterpretation”, unlike “substance use” and “depressive symptoms”, which are mal-adaptive coping mechanisms and represent the lowest average. The variables present a normal distribution, as the skewness and kurtosis values are between −2/+2 and −7/+7. The mechanism “substance use” goes above the skewness and kurtosis values. However, as this is a sample larger than 50 (*n* = 313), this provides statistical robustness for multivariate normal distribution.

The coping mechanisms “active coping”, “planning”, “religion”, “reinterpretation”, “self-blame”, and “acceptance” are positively and significantly correlated with satisfaction with life. In contrast, mechanisms “denial”, “disengagement”, “substance use”, and “depressive symptoms” are negatively and significantly correlated with satisfaction with life. As seen from the data in Table 2, the variables “active coping”, “planning”, “reinterpretation”, “acceptance”, and “satisfaction with life” show a negative and significant correlation with the variable “depressive symptoms”. In contrast, the mechanisms “self-blame”, “denial”, “self-distraction”, “disengagement”, and “substance use” have a positive and significant correlation with depressive symptoms. There does not appear to be a significant correlation between “instrumental support”, “social support”, “religion”, “venting emotions”, and “humor” with the variable “depressive symptoms” or with the mechanisms “instrumental support”, “social support”, “venting emotions”, “self-distraction”, and “humor” with the variable “satisfaction with life”. For more information, consult Table 2.

### 3.2. Structural Equation Modeling Analysis

Table 3 highlights the traditional and incremental adjustment values for the 14 structural equation models. According to the cutoff values, the models “active coping”, religion”, “reinterpretation”, “acceptance”, and “venting emotions” present acceptable tradition and incremental adjustment values for the analyzed models, unlike the models “planning”, “self-blame” “denial”, “self-distraction”, “disengagement”, and “substance use”. However, the obtained values are close to the cutoff values for the adjustment indexes CFI and TLI ≥ 0.90 and SRMR and RMSEA ≤ 0.8 proposed in the literature. In this context, the analysis was conservatively continued for examination of direct and indirect effects.

### 3.3. Direct and Indirect Effects

In Table 4, it is possible to analyse the direct and indirect effects between constructs, namely: (a) the adaptive coping mechanisms “active coping”, “planning”, “reinterpretation”, and “acceptance” present a positive and significant association with satisfaction with life; (b) the maladaptive coping mechanisms “self-blame”, “denial”, “self-distraction”, “disengagement”, and “substance use” present a negative and significant association with satisfaction with life; (c) satisfaction with life is negatively and significantly related to depressive symptoms in the models “active coping”, “planning, “religion”, “reinterpretation”, “acceptance” (adaptive coping mechanisms), “self-blame”, “venting emotions”, “denial”, “self-distraction”, “disengagement”, and “substance use” (maladaptive coping mechanisms).

Regarding the indirect effects between constructs highlighted in Table 5, the coefficients indicate that the adaptive coping mechanisms of “active coping”, “planning”, “reinterpretation”, and “acceptance” have a significant negative, indirect association with depressive symptoms via satisfaction with life. On the other hand, the maladaptive coping mechanisms of “self-blame”, “denial”, “self-distraction”, “disengagement” and “substance use” have a significant positive association with depressive symptoms via satisfaction with life.

## 4. Discussion

The objective of this study was to analyze the mediating role of satisfaction with life between coping mechanisms and depressive symptoms. To fulfil this objective, fourteen structural equation models were analyzed considering the fourteen coping mechanisms as independent variables, the depressive symptoms as dependent variables, and satisfaction with life as a possible mediator. The results are discussed considering current literature.

According to the statistical analysis of the fourteen coping mechanisms, only eleven were able to be assessed regarding direct and indirect effects, with five of those referring to adaptive coping and six referring to maladaptive coping. Three models related to the coping mechanisms “instrumental support”, “social support”, and “humor” were not analyzed, as the models did not present convergence; in other words, the collected data do not support a correct statistical analysis. Therefore, in order to prevent extrapolation of biased data, these models are not discussed below.

Considering the results of this study in terms of direct effects, the coping mechanisms “active coping”, “planning”, “reinterpretation”, and “acceptance” have a significant positive association with satisfaction with life. The maladaptive mechanisms “self-blame”, “denial”, “self-distraction”, “disengagement”, and “substance use” have a significant negative relation with satisfaction with life. These results also reveal a significant negative association between satisfaction with life and depressive symptoms in the models, considering the following coping mechanisms as independent variables: “active coping”, “planning”, “religion”, “reinterpretation”, and “acceptance” [adaptive coping mechanisms] and “self-blame”, “venting emotions”, “denial”, “self-distraction”, “disengagement”, and “substance use” (maladaptive coping mechanisms).

When looking at the results of direct effects, these results are in line with statements of previous theoretical and empirical studies [3,26,27]. Specifically, these studies support these results, as individuals using adaptive coping mechanisms, or problem-focused coping, tend to adapt better to stressful events. The mechanism “active coping”, for example, helps people move towards the elimination of the problem [36], and these people showcase fewer depressive symptoms. On the contrary, individuals using maladaptive coping mechanisms, or emotion-focused coping, are less likely to be able to succeed with stressful situations [26], indicating that those resorting to maladaptive coping mechanisms present more negative indicators of mental health (e.g., depressive symptoms, stress, anxiety). Therefore, adaptive coping mechanisms, or problem-focused coping, can play a key role in reducing depressive symptoms. Furthermore, the study by Tran and Chantagul [3] relates an increased use of adaptive coping mechanisms and problem-focused coping with a higher level of satisfaction with life, while an increased use of maladaptive and emotion-focused coping strategies relate to a lower level of satisfaction with life. In turn, satisfaction with life has a significant negative correlation with depressive symptoms, as described by Moksnes et al. [27] and supported by this study. In fact, current evidence corroborate that satisfaction with life is an important indicator regarding mental health.

The more an individual uses adaptive coping mechanisms, the less depressive symptoms they exhibit as well as depression, anxiety, and stress; opposite to that, those who tend to use maladaptive coping reveal higher levels of depressive symptoms, depression, anxiety, and stress [3,27,36]. The present results indicate that using adaptive coping is more beneficial than using maladaptive coping, reflected by lower levels of depressive symptoms. In other words, this study suggests that individuals can effectively control negative emotions associated with depressive symptoms in day-to-day life by selecting appropriate coping strategies, that is, those focused on adaptive coping to better deal with the situation and maintain ideal mental balance.

Regarding indirect effects, the coefficients indicate that adaptive coping mechanisms have a significant negative and indirect relationship with depressive symptoms when looking at satisfaction with life as a mediator, specifically the mechanisms “active coping”, “planning”, reinterpretation”, and “acceptance”. The maladaptive coping mechanisms “self-blame”, “denial”, “self-distraction”, “disengagement”, and “substance use” present a significant positive and indirect association with depressive symptoms via satisfaction with life.

Indeed, when looking at the analysis of the data referring to indirect effects, these results are in accordance with those of McKnight et al. [26] and Moksnes et al. [22,27] by coming to the realization that satisfaction with life plays a significant mediating role in the relationship between depressive symptoms and coping mechanisms of which the latter are directly and significantly associated with the levels of satisfaction with life [3]. The study by McKnight et al. [26] showed that satisfaction with life mediated the relationship between most stressful events in one’s life and internalization symptoms (e.g., depression). The study by Moksnes et al. [27] showed that satisfaction with life was a partial mediator in the relationship between stress and depressive symptoms. Taking into consideration that there is a significant negative relationship between adaptive coping mechanisms and depressive symptoms, it is confirmed that satisfaction with life acts as a mediator. The results indicate that using adaptive coping, potentiated by satisfaction with life, tends to be associated to lower depressive symptoms. On the contrary, the use of mal-adaptive coping, even when there are high levels of satisfaction with life, tends to be associated to higher levels of depressive symptoms.

The perception of satisfaction with life seems to be adjusted according to the applied coping mechanisms. In particular, the results show that “active coping”, “planning”, “reinterpretation”, and “acceptance” are variables with a significant negative relation with depressive symptoms via satisfaction with life. The use of adaptive coping allows for recovery of well-being and health, thus improving the level of satisfaction with life. Activities that involve planning the future in a way that helps individuals to feel in control of their lives is a way to obtain adaptive coping that leads to higher levels of satisfaction with life and, consequently, less depressive symptoms. The use of appropriate strategies in daily activities allows one to deal with the problems and accept them, either by forgiveness, by controlling hostile thoughts, or managing relationships with others. These behaviors are also associated with the resilience capacity that each individual has in the face of adverse situations in which one does not give into the pressure of the problem regardless of what it is. As for maladaptive coping, it was observed that maladaptive mechanisms that potentiate depressive symptoms are the use of drugs [alcohol, heavy drugs, etc.], denial of what is taking place, blaming oneself for the situation in cause, and constant disengagement day after day. Therefore, “self-blame”, “denial”, “self-distraction”, “disengagement”, and “drug use” are the variables showcasing a significant positive relation with depressive symptoms via satisfaction with life. The higher the persistence of these maladaptive coping mechanisms, the lower the satisfaction with life, as these are behaviors that lead to inner discomfort, and consequently, there is a higher probability of suffering from depressive symptoms or, if they are already present, potential for their worsening.

In general, satisfaction with life mediates the relationship between coping mechanisms and depressive symptoms. The adaptive coping mechanisms and problem-focused mechanisms are positive predictive factors in an increase of the level of satisfaction with life, as they can indicate lower depressive symptoms. Contrary to that, maladaptive coping mechanisms, which are emotion-focused, are negative predictive factors in the reduction of satisfaction with life, as they can indicate higher depressive symptoms. Therefore, the type of coping mechanism typically used by one individual can give an indication of the presence of depressive symptoms.

### 4.1. Limitations

This study showcases important results when it comes to the determinants of the mediating role of satisfaction with life in the relationship between coping mechanisms and depressive symptoms. However, some limitations must be taken into consideration. First, this study had a cross-sectional design, which limits the ability to draw conclusions regarding the causality of the relationships in study. As such, it is pertinent that future studies test the relation between these variables using a longitudinal methodology and, ideally, also experimental in order to verify if the results are consistent. Also, within the sphere of the study methodology, it is also important to note that this study was performed using data from a Portuguese sample. Socio-demographic parameters, such as culture and age, may influence the results [1].

Despite these limitations, this research is highlighted by the empiric study it includes which, up to date, was non-existent in general nor with a representative sample of the Portuguese population. As a suggestion, it would be interesting to analyze the results taking into consideration the age of the participants, as the sample incorporates individuals from 18 to 70 years of age, as well as gender in order to compare male and female results, since literature supports a higher prevalence of depressive symptoms in women [1]. This suggestion stems from the fact that the sample is heterogenous, and we could not fulfil that goal on this study.

### 4.2. Practical Implications

Despite the previous limitations, this study presents contemporaneous evidence and promotes a better knowledge of the connection between satisfaction with life, different coping mechanisms, and depressive symptoms, which is fundamental to develop tools to promote better levels of general well-being and lower levels of negative psychologic indicators [e.g., stress, anxiety].

According to Batista and Oliveira [37], including physical exercise in treatment protocols of depression and depressive symptoms benefits the individual, bringing significant improvement after a few weeks of physical activity (e.g., reducing the symptoms and promoting emotional relief). A way to captivate possible physical exercise enthusiasts is to use dialogue in physical assessments to understand personal preferences and needs, simultaneously working in strategies for behavioral change and adoption of adaptive coping mechanisms, such as proactivity (active coping) and planning. Psychological support consultations and social activities for emotional management can also help with the adoption of adaptive coping mechanisms, as they can allow individuals to work in their expectations, create relaxation routines, and develop new personal strategies to face their issues, leading to reinterpretation or acceptance of the situation [38]. Other strategies to take into consideration are participation in formation and workshops on personal management, organizing and planning daily tasks [in order to understand the true productive uses of time and how one can balance well-being, relations, and professional life], using relaxation techniques, and even reading books on the topic [39,40].

There are certain determinants in the quality of life that can influence the promotion of coping mechanisms, such as mindfulness, which encompasses focusing on the present moment without external or internal judgement [41]. Mindfulness-based interventions reduce stress and promote effective coping [42]; in other words, mindfulness helps in the reduction of maladaptive coping mechanisms. As these interventions are particularly effective in the reduction of stress by reducing repetitive and persistent thoughts [43], it is important to promote them. This promotion may be made by municipal councils, social media marketing, promotional videos, and public support, as examples.

It appears essential that health professionals interested in behavior modification (specifically changing at-risk lifestyles to healthy lifestyles and promoting physical and psychological health) are able to promote the use of adaptive coping mechanisms that are effective in the management of potentially harmful symptoms in the quality of life of the individual [43]. Both mindfulness and support coaching contribute to this ideal of health and well-being, as they facilitate effective coping and increase one’s ability to deal with stressful events in a flexible way, in line with the knowledge that individuals using effective coping strategies report less illness, longer longevity, and better quality of life.

## 5. Conclusions

The study was based on the fact that Portugal is one of the countries with higher prevalence of mental disorders and that, although there are suggestions that coping mechanisms may act as an adaptive tool to depressive state, there are not many studies that analyze the mediating role of satisfaction with life between these and depressive symptoms. To date, no study has analyzed this relationship in the Portuguese population, making this a relevant study useful to assess the relationship between satisfaction with life, the fourteen coping mechanisms, and depressive symptoms.

This study shows that the deeper analysis of the link between different coping mechanisms, satisfaction with life, and depressive symptoms may provide essential knowledge to develop tools to promote more well-being and less negative psychologic indicators (e.g., stress, anxiety). Based on currently available evidence and the analysis in this study, the results demonstrate that, indeed, satisfaction with life is a mediator in the relationship between coping mechanisms used to manage stressful events in one’s life and depressive symptoms.

The current study allows for a deeper understanding of the mediating role of satisfaction with life in the relationship between adaptive and maladaptive coping mechanisms and depressive symptoms. This is also the first study that does so in the Portuguese population, making it more relevant. It therefore suggests that it is important to develop and apply tools to promote the use of adaptive coping mechanisms and reduce maladaptive coping mechanisms. The use of these tools is supported by this study, as with them, health professionals can develop prevention and control strategies using the coping mechanisms investigated on this study and do so in an individual and tailored way.

## Figures and Tables

**Table 1 healthcare-09-00787-t001:** Symptoms of depression.

Manual for the Diagnostic and Statistics of Mental Disorders [7]
Depressed humor: sadness, hopelessness, discouragement, “feeling low” Lack of interest or pleasure in activities that were previously enjoyed Changes in appetite, with significant weight loss or weight gain Insomnia or hypersomnia Psychomotor changes marked by agitation or slowness Fatigue or loss of energy Feelings of guilt or personal devaluation, frequently regarding daily situations Reduced ability to concentrate, think, and make decisions Death thoughts: suicidal ideation

**Table 2 healthcare-09-00787-t002:** Descriptive analysis and correlations between the variables being studied.

Name of the Variables	M	SD	S	K	1	2	3	4	5	6	7	8	9	10	11	12	13	14	15
1. Active coping	3.73	0.84	−0.77	0.90	1														
2. Planning	3.99	0.78	0.96	1.35	0.72 **	1													
3. Instrumental support	3.16	0.94	0.13	0.58	0.25 **	0.35 **	1												
4. Social support	3.19	1.08	0.07	0.78	0.18 **	0.21 **	0.51 **	1											
5. Religion	2.23	1.20	0.76	0.53	0.15 **	0.16 **	0.27 **	0.29 **	1										
6. Reinterpretation	3.50	0.91	0.21	0.31	0.47 **	0.41 **	0.19 **	0.19 **	0.21 **	1									
7. Self-blame	3.36	0.85	0.24	0.04	0.16 **	0.17 **	0.22 **	0.15 **	0.02	0.04	1								
8. Acceptance	3.31	0.86	0.11	0.41	0.28 **	0.33 **	0.24 **	0.20 **	0.12 *	0.36 **	0.14 *	1							
9. Venting emotions	3.25	1.01	0.14	0.21	0.17 **	0.27 **	0.34 **	0.38 **	0.13 *	0.11 *	0.25 **	0.20 **	1						
10. Denial	2.20	0.86	0.48	0.00	−0.12 *	−0.08	0.20 **	0.20 **	0.19 **	−0.01	0.24 **	0.03	0.32 **	1					
11. Self-distraction	3.05	0.94	0.06	0.68	0.04	0.06	0.18 **	0.26 **	0.11 *	0.13 *	0.18 **	0.21 **	0.25 **	0.21 **	1				
12. Disengagement	1.72	0.81	1.17	1.30	−0.33 **	−0.29 **	−0.07	−0.01	0.03	0.17 **	0.12 *	−0.06	−0.03	0.26 **	0.21 **	1			
13. Substance use	1.31	0.62	2.73	9.74	−0.07	−0.09	0.05	0.08	−0.02	−0.03	0.19 **	0.05	0.17 **	0.25 **	0.08	0.27 **	1		
14. Humor	2.86	0.92	0.41	0.21	0.15 *	0.10	0.06	0.03	0.00	0.36 **	0.03	0.29 **	0.08	0.08	0.26 **	0.13 *	0.17 **	1	
15. Satisfaction with life	3.30	0.78	0.30	0.20	0.21 **	0.19 **	0.04	0.04	0.11 *	0.38 **	0.17 **	0.21 **	0.09	−0.17 **	−0.10	−0.24 **	−0.13 *	−0.01	1
16. Depressive symptoms	1.37	0.37	1.37	1.47	−0.25 **	−0.20 **	−0.00	0.11	−0.01	−0.35 **	0.31 **	−0.19 **	0.05	0.18 **	0.15 **	0.32 **	0.24 **	−0.07	−0.63 **

Notes: * *p* < 0.05; ** *p* < 0.01.

**Table 3 healthcare-09-00787-t003:** Adjustment values.

Model	χ^2^	gl	CFI	TLI	SRMR	RMSEA (90% CI)
Active coping	79.423 *	19	0.934	0.903	0.047	0.101 (0.079; 0.124)
Planning	84.934 *	19	0.920	0.882	0.049	0.105 (0.083; 0.129)
Instrumental support	-	-	-	-	-	-
Social support	-	-	-	-	-	-
Religion	82.131 *	19	0.936	0.905	0.045	0.103 (0.061; 0.127)
Reinterpretation	79.666 *	19	0.939	0.910	0.045	0.101 (0.079; 0.125)
Self-blame	108.118 *	19	0.893	0.843	0.056	0.123 (0.101; 0.146)
Acceptance	78.930 *	19	0.932	0.900	0.049	0.101 (0.078; 0.124)
Venting emotions	86.858 *	19	0.933	0.902	0.052	0.107 (0.085; 0.130)
Denial	93.481 *	19	0.919	0.881	0.054	0.112 (0.090; 0.135)
Self-distraction	83.951 *	19	0.930	0.897	0.050	0.105 (0.082; 0.128)
Disengagement	112.619 *	19	0.914	0.874	0.064	0.126 (0.104; 0.149)
Sustance Use	135.904 *	19	0.888	0.836	0.073	0.140 (0.119; 0.163)
Humor	-	-	-	-	-	-

Notes: χ^2^, chi-square test; gl, degrees of freedom; CFI, comparative fit index; TLI, Tucker–Lewis index; SRMR, standardized root mean square residual; RMSE, root mean squared error of approximation; 90% CI, 90% confidence interval of RSMEA; * *p* < 0.001.

**Table 4 healthcare-09-00787-t004:** Direct effects between constructs.

Model	Direct Effect	β	IC 95%
Active coping	Active coping → SL	0.29	0.18; 0.39
	SL → Depressive symptoms	−0.70	−0.76; −0.65
Planning	Planning → SL	0.23	0.07; 0.38
	SL → Depressive symptoms	−0.70	−0.76; −0.65
Religion	Religion → SL	0.10	−0.02; 0.22
	SL → Depressive symptoms	−0.70	−0.75; −0.64
Reinterpretation	Reinterpretation → SL	0.46	0.37; 0.55
	SL → Depressive symptoms	−0.70	−0.76; −0.65
Self-blame	Self-blame → SL	−0.50	−0.71; −0.30
	SL → Depressive symptoms	−7.34	−8.32; −6.37
Acceptance	Acceptance → SL	0.26	0.15; 0.37
	SL → Depressive symptoms	−0.70	−0.76; −0.65
Venting emotions	Venting emotions → SL	0.03	−0.13; 0.20
	SL → Depressive symptoms	−0.70	−0.75; −0.64
Denial	Denial → SL	−0.02	−0.32; −0.01
	SL → Depressive symptoms	−0.70	−0.75; −0.64
Self-distraction	Self-distraction → SL	−0.15	−0.26; −0.04
	SL → Depressive symptoms	−0.70	−0.76; −0.64
Disengagement	Disengagement → SL	−0.31	−0.40; −0.21
	SL → Depressive symptoms	−0–70	−0.76; −0.65
Substance use	Substance Use → SL	−0.15	−0.25; −0.04
	SL → Depressive symptoms	−0.70	−0.76; −0.65

Notes: SL, satisfaction with life.

**Table 5 healthcare-09-00787-t005:** Indirect effects.

Indirect Effect	β	IC 95%
Active coping → SL → Depressive symptoms	−0.20	−0.28; −0.13
Planning → SL → Depressive symptoms	−0.16	−0.27; −0.05
Religion → SL→ Depressive symptoms	0.07	−0.15; 0.01
Reinterpretation → SL→ Depressive symptoms	−0.33	−0.40; −0.26
Self-Blame → SL→ Depressive symptoms	0.18	0.08; 0.28
Acceptance → SL→ Depressive symptoms	−0.18	−0.26; −0.10
Venting emotions → SL→ Depressive symptoms	−0.02	−0.14; 0.09
Denial → SL→ Depressive symptoms	0.15	0.07; 0.22
Self-distraction → SL→ Depressive symptoms	0.11	0.03; 0.18
Disengagement → SL→ Depressive symptoms	0.22	0.14; 0.29
Substance use → SL→ Depressive symptoms	0.10	0.03; 0.18

Notes: SL, satisfaction with life.

## Data Availability

Due to issues of participant consent, data will not be shared publicly. Interested researchers may contact the board from the Research Center (omitted for review) associated in this study.

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
