# Peer review of "Satisfaction with Life: Mediating Role in the Relationship between Depressive Symptoms and Coping Mechanisms"

_healthcare, 2021, doi:10.3390/healthcare9070787_

Round 1

Reviewer 1 Report

This is an interesting manuscript  that aims to find interrelationships between different coping mechanisms and life satisfaction. However some update is still needed.

  1. The introduction section is rather lengthy. It could be better to make it more concise.
  2. According to the results of the manuscript, the role of religion seems to be controversial. It could be better to clarify this point based on the direct and indirect effects of religion.
  3. In the methods section, adding two introductory sentences on the use of the structural equation modeling and why did you use this method could beneficial.
  4. The main findings are not emphasized in the abstract. The abstract  might need to be rewritten to highlight the research question the authors aims to answer. Then it could be a good idea to  show why current knowledge does not answer their question. Following that, it may be better to  describe how the authors solved the research questions  and how they interpret the results.

Author Response

This is an interesting manuscript that aims to find interrelationships between different coping mechanisms and life satisfaction. However, some update is still needed.

R: We appreciate your positive feedback. Substantial revisions were made according to the reviewer's comments.

The introduction section is rather lengthy. It could be better to make it more concise.

R: The introduction section was substantially reduced.

According to the results of the manuscript, the role of religion seems to be controversial. It could be better to clarify this point based on the direct and indirect effects of religion.

R: We appreciate your comment but are unable to understand your statement concerning the “controversy of religion”, based on our results. Could you please help us clarify?

In the methods section, adding two introductory sentences on the use of the structural equation modeling and why did you use this method could beneficial.

R: Sentences were added: “This type of statistical analysis was considered as it capable of producing direct and indirect effects. Additionally, structural equation modeling is a modern statistical method that allows one to evaluate causal hypotheses on a set of intercorrelated nonexperimental data, which is the case of this research”.

The main findings are not emphasized in the abstract. The abstract might need to be rewritten to highlight the research question the authors aims to answer. Then it could be a good idea to show why current knowledge does not answer their question. Following that, it may be better to describe how the authors solved the research questions and how they interpret the results.

R: The abstract was substantially revised.

Reviewer 2 Report

Thank you for an opportunity to review a manuscript presenting a study on the mediating role of satisfaction with life in the relationship between depressive symptoms and coping mechanisms. The authors have compiled an impressive review of the literature and presented the study background, discussion of results and practical implications in much detail. My concern is that it's only too easy to get lost in the detail and not be able to see the forest for the trees. As such, I suggest the authors consider shortening the text of the mss and focusing on the most important information directly pertinent to the study aim and implications.

Materials and methods - can the authors present details of recruitment strategy and the response rate? Were there power calculations to estimate the sample size needed for the statistical analysis? 

I suggest the authors call the study design "cross-sectional" instead of a "transversal study".  

Author Response

Thank you for an opportunity to review a manuscript presenting a study on the mediating role of satisfaction with life in the relationship between depressive symptoms and coping mechanisms. The authors have compiled an impressive review of the literature and presented the study background, discussion of results and practical implications in much detail.

R: We appreciate your positive point of view. Substantial revisions were made in the entire manuscript.

My concern is that it's only too easy to get lost in the detail and not be able to see the forest for the trees. As such, I suggest the authors consider shortening the text of the manuscript and focusing on the most important information directly pertinent to the study aim and implications.

R: The introduction section was substantially reduced.

Materials and methods - can the authors present details of recruitment strategy and the response rate?

R: A non-probabilistic sampling technique to collect data was used, specifically, data was collected from a convenience sample of the population. Unfortunately, we are unable to provide a response rate, since questionnaires were provided to the potential participants using an online questionnaire.

Were there power calculations to estimate the sample size needed for the statistical analysis?

R: Sample size for multivariate analyzes was performed according to the recommendations of Westland (2010), using the online calculator by Soper (2021). We included the following parameters: Anticipated effect size (0.2); Desired statistical power level (0.8); Number of latent variables (2); Number of observed variables (1) and Probability level (0.05). Considering these parameters, the minimum required sample was 223, which was respected in the present study. This information was inserted into the manuscript.

I suggest the authors call the study design "cross-sectional" instead of a "transversal study"

R: The reviewer is right. “Transversal” was changed for “cross-sectional”

Round 2

Reviewer 2 Report

Authors have addressed my queries, thank you.

This manuscript is a resubmission of an earlier submission. The following is a list of the peer review reports and author responses from that submission.